# Chronic Cannabigerol as an Effective Therapeutic for Cisplatin-Induced Neuropathic Pain

**DOI:** 10.3390/ph16101442

**Published:** 2023-10-11

**Authors:** Rahul Nachnani, Diana E. Sepulveda, Jennifer L. Booth, Shouhao Zhou, Nicholas M. Graziane, Wesley M. Raup-Konsavage, Kent E. Vrana

**Affiliations:** 1Department of Pharmacology, Penn State College of Medicine, Hershey, PA 17033, USA; dcastro@pennstatehealth.psu.edu (D.E.S.); ngraziane@pennstatehealth.psu.edu (N.M.G.); kvrana@pennstatehealth.psu.edu (K.E.V.); 2Department of Anesthesiology and Perioperative Medicine, Penn State College of Medicine, Hershey, PA 17033, USA; 3Department of Comparative Medicine, Penn State College of Medicine, Hershey, PA 17033, USA; jbooth@pennstatehealth.psu.edu; 4Division of Biostatistics and Bioinformatics, Penn State College of Medicine, Hershey, PA 17033, USA; szhou1@pennstatehealth.psu.edu

**Keywords:** cannabinoids, neuropathy, CIPN, cisplatin, dorsal root ganglia, analgesia, sex differences, tolerance

## Abstract

Cannabigerol (CBG), derived from the cannabis plant, acts as an acute analgesic in a model of cisplatin-induced peripheral neuropathy (CIPN) in mice. There are no curative, long-lasting treatments for CIPN available to humans. We investigated the ability of chronic CBG to alleviate mechanical hypersensitivity due to CIPN in mice by measuring responses to 7 and 14 days of daily CBG. We found that CBG treatment (i.p.) for 7 and 14 consecutive days significantly reduced mechanical hypersensitivity in male and female mice with CIPN and reduced pain sensitivity up to 60–70% of baseline levels (*p* < 0.001 for all), 24 h after the last injection. Additionally, we found that daily treatment with CBG did not evoke tolerance and did not incur significant weight change or adverse events. The efficacy of CBG was independent of the estrous cycle phase. Therefore, chronic CBG administration can provide at least 24 h of antinociceptive effect in mice. These findings support the study of CBG as a long-lasting neuropathic pain therapy, which acts without tolerance in both males and females.

## 1. Introduction

Over forty years have passed since the first use of platinum compounds to treat a variety of cancers, including testicular, ovarian, lung, and breast cancers [1]. Drugs, like cisplatin, have been used alone and in concert with other agents to induce DNA adducts and slow the proliferation of cancer cells. While barriers to cisplatin success such as cancer resistance mechanisms and nephrotoxicity have been studied and interventions implemented—an important, dose-limiting, adverse effect of platinum therapy has not yet been feasibly addressed: cisplatin-induced peripheral neuropathy (CIPN).

The sole pharmacological treatment for chemotherapy-induced neuropathy with clinical trial evidence remains duloxetine, which proves incomplete in patients due to accumulated tolerance, side effects, and variable efficacy [2]. Additionally, CIPN may escalate over time without additional exposure to cisplatin, and pain persists for months to years after one’s last dose of cisplatin [3]. Patients, providers, and caregivers often turn to other medications for various pain indications such as gabapentin, tricyclic antidepressants, natural products, or medicated topicals, to ameliorate the chronic and debilitating nature of the pain; however, none of these have successfully shown benefits in clinical trials [4]. There is growing patient and research interest in cannabinoid formulations with Δ9-tetrahydrocannabinol (THC) for pain syndromes, in part due to wide pharmacological activity with relatively limited side effects. While these pain therapeutics appear effective, recent clinical data suggest that patients may increase use over time to maintain analgesic efficacy, raising concerns for tolerance and abuse liability [5].

Novel non-euphoric cannabinoids, like cannabigerol (CBG), are concurrently gaining traction due to improved synthesis methods and changing public perception [6]. In fact, many recent studies of CBG in rodent models as well as human survey data suggest an increasing interest in using the compound to address human disease [7,8,9,10,11,12].

The varied and unique pharmacodynamic profile of CBG provides evidence for its analgesic potential [6]. Recently, we demonstrated that an acute injection of CBG significantly reduced mechanical hypersensitivity in a mouse model of CIPN partially through adrenergic and cannabinoid receptors [10]. Here, we extended these findings to investigate the effects of chronic CBG treatment on neuropathic pain using a treatment paradigm that is more likely to be encountered in the patient population. Therefore, both male and female mice with CIPN were treated daily with CBG allowing us to test the hypothesis that chronic CBG treatment would provide a lasting reduction in neuropathic pain without eliciting tolerance to CBG. Finally, because gene expression in the mouse dorsal root ganglia (DRG) is sensitive to cisplatin [13], we investigated the effects of chronic CBG on gene expression changes in lumbar DRG of CIPN mice, compared to vehicle mice in lumbar DRG.

## 2. Results

A visual timeline of experimental procedures is provided in Figure 1.

### 2.1. Daily Cannabigerol Treatment Does Not Affect Weight or Induce Adverse Events in Neuropathic Mice

We observed that pure cannabigerol (administered i.p.) at 10 mg/kg for males and 15 mg/kg for females did not affect weight changes (males: 0.02 g/day, *p* = 0.318 and females: 0.09 g/day, *p* = 0.603) (Table 1). Weight changes over individual timepoints can be found in Appendix A. None of the mice experienced any of the following adverse events: diarrhea, injection site reactions, hematuria, or skin lesions. None of the mice died during treatment or during induction of cisplatin-induced neuropathy.

### 2.2. CBG Relieves CIPN Mechanical Hypersensitivity and Does Not Vary Based on Estrous Cycle Phase

We have previously published [10] findings that acute CBG relieves mechanical hypersensitivity in neuropathic pain for up to six hours. The present study replicated the pain relief at 1 h on the first day of daily treatments. Male and female mice treated with 10 mg/kg CBG or 15 mg/kg of CBG, respectively, had significantly higher acute pain relief compared to the post-neuropathy readings (Figure 2, *p* < 0.0001 for both male and female mice). As noted in the Discussion and previously published work, female mice require higher doses to achieve similar analgesic effects in this model. Importantly, using the baseline force (g) as 100% pain tolerance, male mice returned to 85.7% ± 17.5% and female mice returned to 80.0% ± 12.4% of naïve pain tolerance (mean ± SD) (Appendix A). In males, acute CBG injection reversed the effects of CIPN to baseline values (*p* > 0.05, 2-way ANOVA). In female mice, acute 15 mg/kg CBG injection attenuated, but did not completely reverse mechanical hypersensitivity due to CIPN (*p* < 0.05, 2-way ANOVA).

We also previously reported varying dose-response relationships in females following acute CBG [10,14]. To understand if the variation in response may be attributed to the estrous cycle phase, we report the individual cycle phase of each female mouse tested and the mouse’s von Frey score. While the female mice receiving CBG were more varied in response to mechanical hypersensitivity in individual cycle phases, there were no associations with the estrous phase (Figure 3).

To investigate the effects of chronic CBG in a mouse model of CIPN, we administered daily injections of CBG for seven and fourteen consecutive days. Von Frey measurements and vaginal lavage were performed before administration of any CBG or vehicle, to prevent capturing acute effects, and 24 h after the most recent injection of CBG or vehicle. Through within group analysis, we found that injections of CBG produced significant analgesia in neuropathic mice, 24 h after the seventh (*p* = 0.0004 for males and *p* < 0.0001 for females) and fourteenth injections (*p* = 0.0006 for males and *p* = 0.0007 for females), respectively (Figure 4 and Figure 5a).

From between group assessments, we found a significant reduction in mechanical hypersensitivity in males treated with CBG compared to vehicle after both 7 and 14 days. Twenty-four hours after seven days of daily injections, males receiving 10 mg/kg CBG scored 5.46 g ± 1.41 g while males receiving vehicle scored 2.01 g ± 0.33 g (mean difference: −3.45 g, 95% confidence interval (CI): (−4.53 g, −2.37 g), *p* < 0.0001). Similarly, females receiving seven days of daily 15 mg/kg CBG injections scored 3.89 g ± 0.59 g while females receiving vehicle injections scored 2.15 g ± 0.74 g (mean difference: −1.74 g, 95% CI: (−2.82 g, −0.66 g), *p* = 0.0003) 24 h after their seventh injection. After fourteen days of daily injections, males receiving daily CBG scored 4.90 g ± 1.15 g while males receiving vehicle scored 2.27 g ± 0.44 g 24 h after their fourteenth injection (mean difference: −2.63 g, 95% CI: (−3.71, −1.55 g), *p* < 0.0001). Females receiving daily injections of CBG scored 4.44 g ± 1.14 g while female mice receiving vehicle scored 2.10 g ± 0.36 g (mean difference: −2.34 g, 95% CI: (−3.42 g, −1.26 g), *p* < 0.0001) 24 h after their fourteenth injection.

In males, 7 days of daily CBG attenuated mechanical hypersensitivity (mean difference: −1.684 g, 95% CI: (−4.06 g, 0.69 g), *p* = 0.19), and 14 days of daily CBG attenuated mechanical hypersensitivity compared to baseline (mean difference: −2.25 g, 95% CI: (−4.01 g, −0.49 g), *p* = 0.014). In females, 7 days of daily CBG attenuated mechanical hypersensitivity compared to baseline (mean difference: −3.79 g, 95% CI: (−4.86 g, −2.72 g), *p* < 0.001) and 14 days of daily CBG attenuated mechanical hypersensitivity compared to baseline (mean difference: −3.24 g, 95% CI: (−4.89 g, −1.58 g), *p* < 0.001)

Using a baseline-corrected analysis, we set each individual mouse’s baseline von Frey rating as 100% to understand how mechanical hypersensitivity within the subject changed over time through this schedule. After induction of neuropathy, male and female mice dropped to about 32.0% (mean, *n* = 20) and 29.0% (mean, *n* = 20) of their baseline pain sensitivity, respectively. After seven daily injections of CBG, male mice returned to 80.6 ± 31.2% of their baseline pain sensitivity, and female mice returned to 51.5 ± 11.1% of their original pain sensitivity measured 24 h after the last injection of CBG. After fourteen daily injections, males returned to 70.9 ± 21.5% of their pain sensitivity and females returned to 59.6 ± 22.1% of their original pain sensitivity, measured 24 h after the last injection of CBG (mean ± SD, *n* = 10 for each group). Graphs of these data can be found in Appendix A. As demonstrated in Appendix A, there is considerable inter-animal variability in the therapeutic response.

Finally, we assessed the estrous cycle phase as described above, to understand if a component of variability among female analgesia levels was due to varying estrous cycle phase. The treatment effect remained significant (after 7 days of injections: *p* < 0.0001; after 14 days of injections: *p* < 0.0001) after considering estrous cycle phase. There does not appear to be any effect of estrous cycle phase on the effect of CBG versus vehicle for pain (after 7 days of injections: *p* = 0.635 (Figure 5b); After 14 days of injections: *p* = 0.655.) (Figure 5c).

### 2.3. Gene Expression Changes from Daily Administration of Cannabigerol in a Selected Panel of Cannabinoid and Pain-Related Targets

Thus far, we have demonstrated that chronic administration of CBG significantly reduces CIPN in male and female mice without apparent tolerance over time. Next, we sought to identify CIPN-sensitive genes in the dorsal root ganglia (DRG) that are modified by this treatment regimen. We analyzed genes known to be targeted by cannabigerol and other cannabinoids, including *Cnr1*, *Cnr2*, *Gpr55*, *Faah*, *Mgll*, *Adra2a-c*, *Pparg* [15,16,17,18,19], or to have been previously published as volatile in a cisplatin-induced neuropathy setting in DRG, including *Drd2*, *Gfap*, and *Oprm1* [13]. Percent differences in relative gene expression (measured by qRT-PCR) between CBG and vehicle groups and statistical significance are reported in Table 2; raw data are provided in Appendix A. In male mice, we identified a 17% decrease in expression of *Atf3* in mice receiving CBG compared to those receiving vehicle (*p* = 0.043). We did not identify a significant difference in *Atf3* in female mice; however, we did identify a decrease in *Drd2* and *Oprm1* expression in female mice, but not in male mice. Female mice receiving daily CBG experienced a 19% reduction in *Drd2* expression (*p* = 0.029), and 9.5% reduction in *Oprm1* expression. No other gene expression differences were significant (*p* > 0.05).

## 3. Discussion

We herein demonstrate that CBG reduces neuropathic pain in a mouse model of CIPN in male and female mice, without development of tolerance or need for dosing more than once a day, regardless of the estrous cycle phase. Chronic administration of CBG surpasses the pharmacokinetic limitations of acute CBG administration for neuropathy, which wears off in about six hours after acute injection, since we observed 24 h reduction in pain following 7 or 14 days of treatment [10]. Additionally, previously published pharmacokinetic data of CBG administered at 120 mg/kg i.p. in mice reported an elimination half-life of slightly under 3 h [20]. Furthermore, we found that the estrous cycle did not play a significant role in modulating the mechanical hypersensitivity responses to CBG analgesia in female mice. Finally, in our model of CIPN, chronic CBG administration did not significantly change gene expression in dorsal root ganglia for many pain- and cannabinoid-relevant genes.

Preliminary safety data are strong, as we identified no mortality, adverse events, or weight changes in our three independent replicates of these findings. Our previous work demonstrated a hypotensive effect from acute administration of CBG [21], likely due to alpha-2 agonist activity, which may be a barrier in clinical use. From a recent survey of recreational CBG users, there were no reports of lightheadedness, fainting, or other hypotensive symptoms, although blood pressure measurements were not recorded as part of the study [8].

Our results showcase no reduction in efficacy after seven and fourteen daily injections of the same dose of CBG in both male and female mice. A deficit in neuropathic pain literature exists for pharmacotherapeutics which maintain efficacy over longer durations of time for chronic illnesses. The strengths of this model of CIPN include the long-lasting neuropathy experienced by mice after four weekly injections of cisplatin; some mice retained neuropathic mechanical hypersensitivity for 2–3 months after completion of cisplatin injections. Other reports, using this model, showcase tolerance to the analgesic effects of THC that developed quickly after daily administration, with female mice demonstrating tolerance to analgesia more quickly than males [22]. In this study, all mice treated with vehicle retained similar pain sensitivity for the entire treatment schedule while mice receiving daily injections of CBG approached baseline levels of pain sensitivity. These results give hope for a more translatable method of pain relief for human conditions which are unrelenting and are currently limited by treatment efficacy tolerance.

Chronic pain syndromes and suffering are more common in women than in men [23]. Previous work studying the effects of menstrual cycle changes of estrogen on pain perception yielded mixed results in both clinical and preclinical settings. While some clinical studies report associations of low estrogen with higher pain sensitivities [24], others report no effects of estrogen and progesterone levels with pain sensitivities [25]. Importantly, the antinociceptive effects of THC were reported to be greater during some stages of the rodent estrous cycle than other stages [26]. Our findings with chronic CBG with respect to the estrous cycle do not support the hypothesis that the cycle phase contributes to analgesic variability of CBG. Rather, our findings support CBG as an effective pain therapeutic in gonadally-intact male and female mice regardless of cycle phase. While responses in female mice were not related to estrous cycle phase, the pharmacological sex differences in response are yet to be explained. Several mechanisms may contribute to differential response including sex differences in CBG metabolism, effects of sex hormones, and differential receptor signaling and density. Indeed, sex differences in human response to cannabinoids is a field of growing interest [27,28,29,30]. Our future work includes pharmacokinetic analysis of metabolic rates and a more thorough understanding of sex hormone effect on analgesic efficacy by correlating serum sex hormone levels to analgesic effect.

The dorsal root ganglia are integral to the development of neuropathic pain and are an important conductor of pain signaling between the peripheral and central nervous system. Because the cell bodies of these expansive pseudo-unipolar neurons are clustered in the ganglia, transcriptional analysis may provide insight into foundations of pain signaling in neuropathy. Unfortunately, the landscape of measuring gene expression in models of neuropathic pain is highly heterogenous. While some reports measure transcriptional changes directly after administration of chemotherapy [13], others may measure changes after addition of a therapeutic at various timepoints [31]. Recent reports of global DRG gene expression changes suggest cisplatin alters activation of inflammatory and neuronal genes [13], as well as altering pain-signaling channels such as *Trpv1* [32]. Investigation of many of these major genes provided few significant differences. We identified modest significant decreases between *Atf3* (Activating Transcription Factor 3) in CBG treated male mice compared to vehicle treated male mice, and *Drd2* (Dopamine Receptor D2) and *Oprm1* (Mu Opioid Receptor 1) in CBG treated female mice versus vehicle treated female mice. All decreases were below 20%, between treatment groups, so the overarching relevance is limited. *Atf3* is a neuronal health marker implicated in several chronic neuropathy models, although it is still unclear whether the gene is helpful or harmful in repairing nervous system damage [33]. *Drd2* and *Oprm1* activation are implicated in pain syndromes, and reduced expression of the receptors may be a result of altered signaling from chronic CBG [34,35].

A major limitation of this work is the lack of a dose-response model for males and females receiving daily doses of CBG. Future work addressing this pharmacological question may reveal additional insight into minimum effective doses and sex differences in responses. The pharmacodynamic profile of CBG is not fully understood, such as its potential interactions with subtypes of the alpha-2 receptor and downstream behavioral effects of alpha-2 activation. Further work must be performed to explore if (1) the hypotensive effect of acute CBG is present in humans, (2) if the hypotensive effect is sustained after chronic use, and (3) if the agonist effect is creating a sedative or anxiolytic effect which confounds antinociceptive effects. The alpha-2 receptor driven hypotensive effect of CBG may be similar to that of the clinically used alpha-2 receptor agonist clonidine, which is effective at reducing blood pressure for emergent and urgent hypertensive crises, but has little-to-no efficacy at reducing blood pressure long term [36]. Additionally, this model of neuropathy only utilizes cisplatin, and other chemotherapy-induced neuropathy models may yield additional insights into the antinociceptive effects of CBG. Finally, the majority of non-significant results of our transcriptional inquiries were surprising but rational, considering the timespan of mRNA and transcriptional changes. Our capturing of gene expression at this stage of the neuropathy and treatment modality (24 h after the last drug injection, 3 weeks after the last cisplatin injection) may not be ideal for transient gene expression changes, despite the marked behavioral difference. Moreover, there is no compelling literature suggesting that pharmacological analgesia should produce changes in gene expression. Our future work will consider protein-level changes as well as circulating endocannabinoids as potential mechanistic contributions to the long-lasting analgesia induce by CBG.

## 4. Materials and Methods

### 4.1. Animals

All experiments were performed in accordance with procedures approved by the Pennsylvania State University College of Medicine Institutional Animal Care and Use Committee (approval number 202001327 and 202202238). Wild-type male (*n* = 20) and female (*n* = 20) age-matched (between 11–13 weeks) C57BL/6J mice (The Jackson Laboratory, Bar Harbor, ME, USA) were group-housed on a twelve-hour light/dark cycle with ad libitum food and water. As noted throughout the manuscript, aspects of findings were replicated in 1–2 additional independent cohorts.

### 4.2. Cisplatin-Induced Neuropathy

The methods for inducing of cisplatin-induced neuropathy in this study are the same as previously described [10,37]. Briefly, male and female mice (*n* = 40 total) were injected with 5 mg/kg intraperitoneal (i.p.) cisplatin (Acros Organics, Fairlawn, NJ, USA), and 4% sodium bicarbonate subcutaneously (s.c.) once a week for four weeks. Sodium bicarbonate was administered to prevent the nephrotoxicity of cisplatin.

### 4.3. Measurement of Mechanical Hypersensitivity—Von Frey

Mechanical sensitivity measurements were taken before the beginning of cisplatin administration and after four weeks of cisplatin, termed “Baseline” and “Neuropathy”, respectively. Measurements were performed by using an electronic von Frey anesthesiometer (IITC Life Sciences Inc., Woodland Hills, CA, USA) exactly as previously described [10,11,14,22,38]. The experimenter performing von Frey was blinded to all treatment groups.

### 4.4. Drug Treatment Schedule and Analgesic Testing

After induction of neuropathy, mice were randomized to one of two treatment arms: vehicle (DMSO, Tween 80, saline [1:1:18]) or cannabigerol (Cayman Chemical (Ann Arbor, MI, USA) CBG, 10 mg/kg for males, 15 mg/kg for females), stratified by sex. Dosing for males mimics previous work [10]. Knowing that in previous work females require more than males, we tested 15 mg/kg females. Mice received daily i.p. injections of their designated treatment (CBG or vehicle) every day at the same time (10:00 h) for 14 days. On the first day of daily injections, von Frey measurements of all mice and vaginal lavage of female mice were performed one hour after injection (11:00 h) to identify acute response to CBG or vehicle. The von Frey measurements of all mice and vaginal lavage of female mice were again performed after seven and fourteen daily injections, 24 h after the last treatment injection. Importantly, to identify chronic effects and avoid acute effects of the drug, von Frey measurements and vaginal lavage were performed before any injections of CBG or vehicle (before 10:00 h), and both procedures were performed within 1 h of each other. Von Frey measurements and vaginal lavage after seven and fourteen daily injections occurred 23–24 h (between 09:00 h and 10:00 h) after each mouse had received its last injection of CBG or vehicle. A visual timeline of experimental procedures is provided in Figure 1. Independent replicates of experiments followed similar timing and dosing protocols (two independent replicates for males, one independent replicate for females).

### 4.5. Estrous Cycle Staging and Cytology 

Estrous cycle stage was identified through cytology of vaginal lavage 30 min after von Frey. Up to 75 μL of normal saline (0.9% NaCl) was used to gently lavage the vaginal opening, and the solution was dispensed onto a glass microscope slide. Once dried, the samples were fixed with ethanol, stained with eosin and methylene blue using the Ephredia™ Shandon™ Kwik-Diff™ staining kit (Catalog (Cat.) #9990700, Fisher Scientific, Pittsburgh, PA, USA), and analyzed microscopically to determine the stage of estrous using methods previously described [39]. In brief, the samples were determined to represent proestrus, estrus, metestrus, or diestrus based upon the cell types present and cell density of the sample on the slide. 

### 4.6. Dorsal Root Ganglia Extraction

Mice were sacrificed 24 h after fourteen days of daily injections. Before sacrifice, mice underwent von Frey measurements 24 h after their last drug injection (cannabigerol or vehicle), and then were sacrificed using isoflurane and cervical decapitation to avoid damage to the spinal cord. Dorsal root ganglia (DRG) extraction in mice was adapted from two previously published protocols [40,41]. Briefly, all procedures were performed on ice and the spinal column was removed from the dorsal aspect of the mouse. After hydraulic excision of the spinal cord with ice-cold Gibco™ Hank’s balanced salt solution (HBSS) (ThermoFisher, Waltham, MA, USA, Cat. #14175095), the vertebral column was placed in a dish. A light microscope was then used to aid in dissection of the lumbar level L3, L4, and L5 DRG, bilaterally. Finally, after removing the neuronal processes on either side of DRG, the cells were placed in 300 μL of RNALater™ stabilization solution (Cat. #RO901, Sigma-Aldrich, Inc., St. Louis, MO, USA) and stored in accordance with the manufacturer’s instructions for further downstream RNA analysis.

### 4.7. RNA Extraction and RT-PCR

All samples underwent RNA isolation using a NucleoSpin RNA Plus, Mini kit (Cat. #740984.50, Macherey-Nagel, Düren, Germany) according to the manufacturer’s instructions. RNA Integrity Number (RIN) was calculated using Agilent 2100 BioAnalyzer (Agilent Technologies, Palo Alto, CA, USA) and resulted in RIN integrity values between 4.0–7.7, adequate for qRT-PCR analysis [42]. An amount of 1000 ng of isolated RNA was reverse transcribed to cDNA using Applied Biosystems High-Capacity cDNA Reverse Transcription Kit with RNase Inhibitor (Cat. #4374966, Applied Biosystems, Foster City, CA, USA). Quantitative real-time polymerase chain reaction (qPCR) was performed in 394-well plates using TaqMan gene expression assays (Cat. #4331182, Applied Biosystems, Foster City, CA, USA), TaqMan Gene Expression Master Mix (Cat. #4369016, Applied Biosystems, Foster City, CA, USA), and RNase-free water. A list of gene expression assays can be found in Appendix A. All assays included a no-reverse-transcriptase control and a no-cDNA control. qPCR was performed using a QuantStudio 12K Flex Real-Time PCR System (Applied Biosystems, Foster City, CA, USA). Analysis of relative quantification of mRNA in samples was performed using the ΔΔCt method, normalized to a housekeeping gene, β-actin [43], and the mean of the vehicle treated group was normalized to “1”. The most appropriate housekeeping gene was determined through the lowest coefficient of variability following a head-to-head comparison of β-actin, GAPDH, and 18S RNA for DRGs in this sample set and for those of an independent experiment.

### 4.8. Data Analysis

GraphPad PRISM software version 9.0.2 (134) and statistical software R version 4.2.2 with package lme4 [44] was used for behavioral and qRT-PCR analyses. All results are presented as mean ± standard deviation with individual datapoints provided in all graphs. For behavioral data, statistical significance (*p* < 0.05) was assessed through two-way ANOVA with a Tukey post hoc test. The difference in weight changes between two treatment arms was assessed using a linear mixed effects regression model. Estrous cycle staging comparisons were calculated using linear regression with F-test. qPCR comparisons were performed using the Mann–Whitney test.

## 5. Conclusions

As the therapeutic potential of cannabigerol gains popularity in the research and public sectors, in-depth characterization of its benefits and harms needs to be conducted. This report is the first of its kind to identify the chronic analgesic potential of cannabigerol in a translational model of neuropathy and demonstrate pharmacokinetic 24/7 relief of pain, without tolerance, and resistant to the effects of the estrous cycle. This novel approach to understanding the aggregate analgesic effects of cannabinoids may be used in other pharmacologic settings, and we are continuing to study the mechanism of cannabigerol as a neuropathic pain reliever through further pharmacodynamic and pharmacometabolic testing, along with synergistic effects with other non-euphoric cannabinoids with analgesic potential.

## Figures and Tables

**Figure 1 pharmaceuticals-16-01442-f001:**
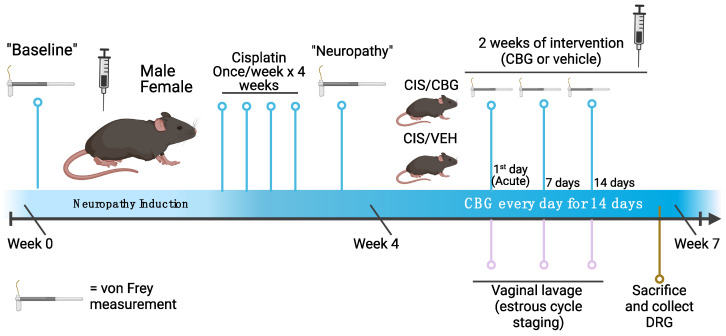
Timeline of experimental methods, behavioral data collection, and sample collection. Created with BioRender.com. CIS = cisplatin, CBG = cannabigerol, VEH = vehicle, DRG = dorsal root ganglia.

**Figure 2 pharmaceuticals-16-01442-f002:**
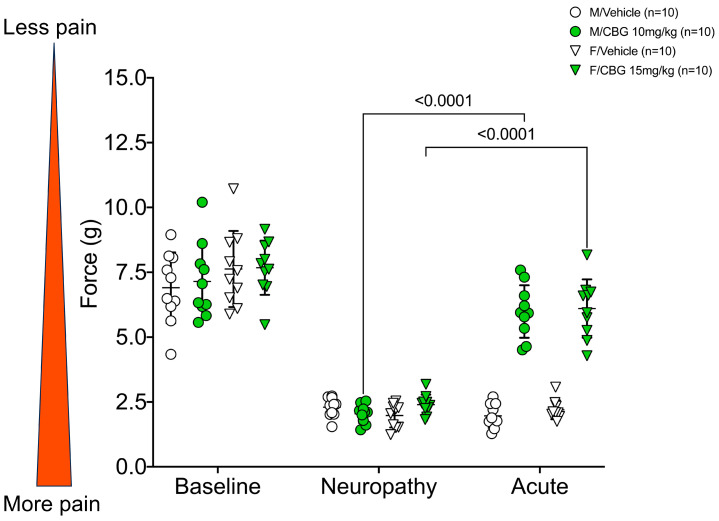
Acute effects of cannabigerol on cisplatin-induced mechanical hypersensitivity in males (*n* = 10 for vehicle and CBG) and females (*n* = 10 for vehicle and CBG). Von Frey measurements measured 1 h after injection on the first day of injections. 2-way ANOVA with Tukey’s multiple comparisons test. M/Vehicle = males receiving vehicle; M/CBG = males receiving cannabigerol; F/Vehicle = females receiving vehicle; F/CBG = females receiving cannabigerol.

**Figure 3 pharmaceuticals-16-01442-f003:**
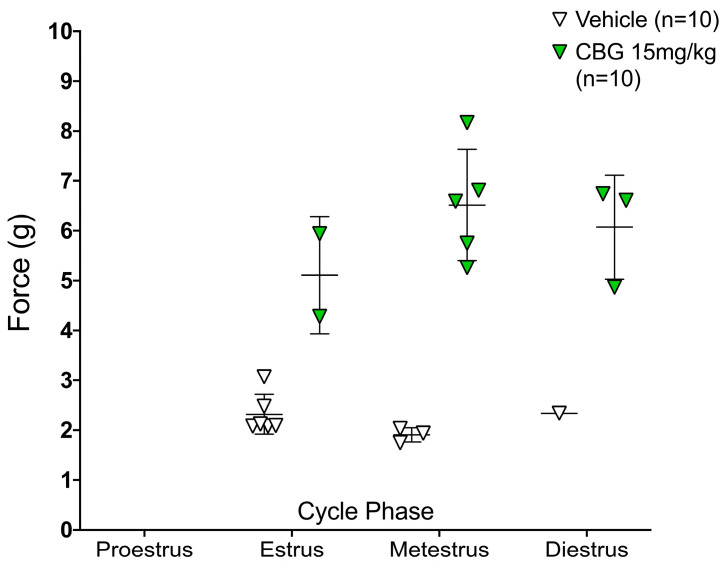
Estrous cycle phase after acute injection of CBG (*n* = 10) or vehicle (*n* = 10). Phase identified one hour after injection of CBG or vehicle on the first day of treatment schedule, graphed versus each female mouse’s mechanical hypersensitivity (von Frey, force in grams).

**Figure 4 pharmaceuticals-16-01442-f004:**
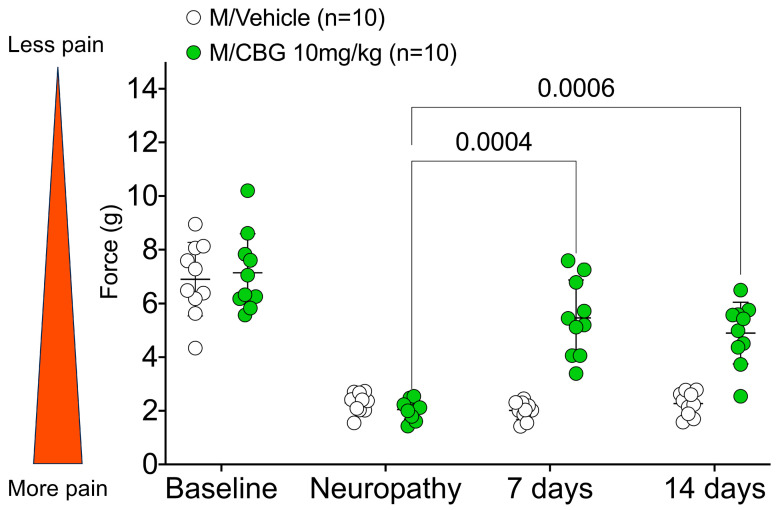
Chronic effects of CBG on neuropathic pain in male mice (*n* = 10 for vehicle and CBG). Force (g) measured 24 h after the last injection. All statistical analyses were conducted using 2-way ANOVA with Tukey’s multiple comparisons test.

**Figure 5 pharmaceuticals-16-01442-f005:**
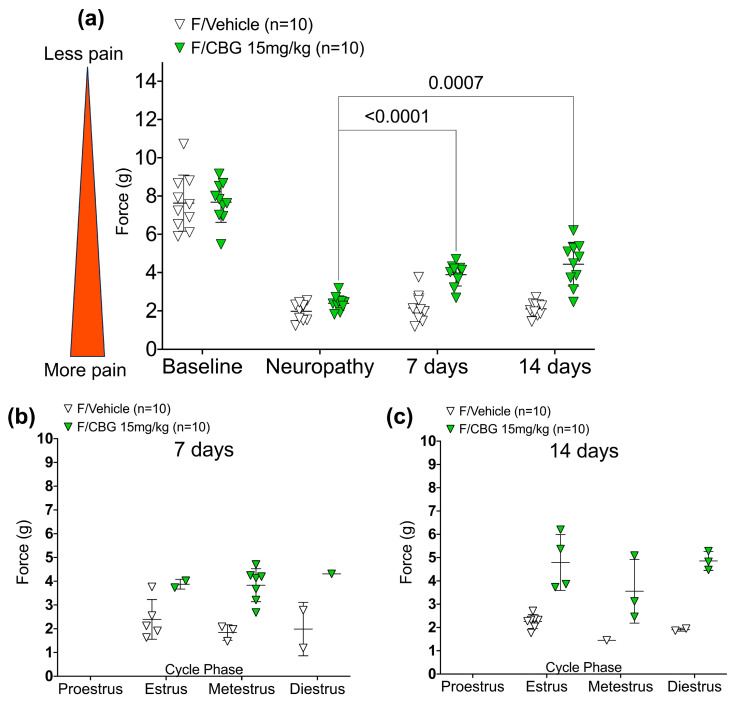
(**a**) Chronic effects of CBG on neuropathic pain in female mice (*n* = 10 for vehicle and CBG). Mechanical hypersensitivity (force (g)) measured 24 h after the last injection. All statistical analyses were conducted using 2-way ANOVA with Tukey’s multiple comparisons test. (**b**) Estrous cycle phase identified twenty-four hours after seventh injection of CBG or vehicle, and within 30 min of the von Frey measurement. (**c**) Estrous cycle phase identified twenty-four hours after fourteenth injection of CBG or vehicle, and within 30 min of the von Frey measurement.

**Table 1 pharmaceuticals-16-01442-t001:** Mean weights of neuropathic mice (*n* = 10 for CBG and vehicle for both males and females) with standard deviation. Measurements displayed at 3 relevant timepoints: Start of daily dosing, after 7 days of daily doses, and after 14 days of daily doses. CBG = cannabigerol, SD = standard deviation. Weight changes over entire timespan of treatment shown in Appendix A.

	Males				Females			
	CBG (*n* = 10)		Vehicle (*n* = 10)		CBG (*n* = 10)		Vehicle (*n* = 10)	
	Mean (g)	SD (g)	Mean (g)	SD (g)	Mean (g)	SD (g)	Mean (g)	SD (g)
Start	27.1	0.985	27.8	0.969	19.8	0.909	20.3	1.38
7 days	27.0	1.17	27.6	0.7	20.2	1.01	20.5	1.59
14 days	28.0	1.24	28.5	0.863	21.1	1.16	21.4	1.19

**Table 2 pharmaceuticals-16-01442-t002:** qRT-PCR analysis of gene expression differences in lumbar dorsal root ganglia (L3–L5) between cannabigerol-treated and vehicle-treated mice.

		Males ^1^		Females ^1^	
Gene	Name	% Change in Expression (CBG/Vehicle)	*p* Value	% Change in Expression (CBG/Vehicle)	*p* Value
*Cnr1*	Cannabinoid Receptor 1	5%	0.684	−5%	0.631
*Cnr2*	Cannabinoid Receptor 2	−38%	0.46	3%	0.631
*Gpr55*	G-protein coupled receptor 55	−10%	0.661	18%	0.579
*Faah*	Fatty Acid Amide Hydrolase	10%	0.28	−1%	0.481
*Mgll*	Monoglyceride Lipase	15%	0.105	2%	0.999
*Atf3*	Activating Transcription Factor 3	−17%	0.043 *	−4%	0.971
*Trpv1*	Transient Receptor Potential Cation Channel Subfamily V Member 1	−1%	0.853	−6%	0.28
*Adra2a*	Adrenergic Receptor 2A	−7%	0.356	−10%	0.796
*Adra2b*	Adrenergic Receptor 2B	−4%	0.661	−31%	0.796
*Adra2c*	Adrenergic Receptor 2C	3%	0.912	7%	0.912
*Drd2*	Dopamine Receptor D2	5%	0.661	−19%	0.029 *
*Gfap*	Glial Fibrillary Acidic Protein	−5%	0.999	−59%	0.684
*Oprm1*	Mu Opioid Receptor 1	7%	0.166	−9.5%	0.007 *
*Pparg*	Peroxisome Proliferator Activated Receptor Gamma	68%	0.321	48%	0.258

^1^ Relative Quantification was calculated using the ΔΔCt method, normalized to β-actin as housekeeping gene. CBG, cannabigerol. Statistical *p* value calculated using Mann–Whitney test, * = *p* value reached significance of ≤0.05.

## Data Availability

All data supporting the findings of this study are available within the paper and its Appendix A.

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
