# Peer review of "Chronic Cannabigerol as an Effective Therapeutic for Cisplatin-Induced Neuropathic Pain"

_pharmaceuticals, 2023, doi:10.3390/ph16101442_

Round 1
Reviewer 1 Report
Dear authors of the manuscript entitled "Chronic cannabigerol as an effective therapeutic for cisplatin-induced neuropathic pain"
It is an interesting work, congratulations!
However, I have some points that you might take into consideration
1- The authors discussed about CBG acting on alpha-2-receptor, but none experiments you showed that the dose used does not provoke sedation. Consedering that alpha-2-receptor agonists might promote this kind effect and mask antinociceptive activity. Did the authors think to evaluate the sedation in the open field to check a possible sedative activity of the CBG dose that it was carried out in the paper and rule out the hypothesis of sedation.
2- I am not convinced why the authors increased the dose to 15 mg/kg i.p. in females. The reference 10 reports that CBG (10 and 20 mg/kg) significantly reduced mechanical hypersensitivity in neuropathic female mice with the 20 mg/kg dose of CBG showing the greatest significant difference compared to vehicle controls. Why the 20 mg/kg dose did not used in the paper?
Minor reviews
3- Replace 10mg/kg and 15 mg/kg to 10 mg/kg and 15 mg/kg
4- Line 184 are is duplicated
5- Figure 2 might be replaced by a table showinf initial weight (before treatment), 7 and 14 days
6 - In the Abstract, replace analgesic relief to antinociceptive effect
The quality of English is good, it requires some occasional adjustments througout the text, please you should read carefully.
Author Response
Thank you very much for the kind words and important feedback to this manuscript. I have uploaded a track changes version of the manuscript to address these changes, and the steps we have taken are outlined in this response.
Comment 1: The authors discussed about CBG acting on alpha-2-receptor, but none experiments you showed that the dose used does not provoke sedation. Consedering that alpha-2-receptor agonists might promote this kind effect and mask antinociceptive activity. Did the authors think to evaluate the sedation in the open field to check a possible sedative activity of the CBG dose that it was carried out in the paper and rule out the hypothesis of sedation.
Response to Comment 1: Thank you for this important consideration. Our collaborators recently published that CBG did not evoke sedation, as measured by locomotor activity, for 3 hours after injection (Vernail et al 2022, Frontiers in Physiology). We therefore did not measure sedation and anxiety behaviors in this study but will expand the scope to include these behavioral metrics in future studies. We have added this to last paragraph in Discussion, on limitations, as follows:
Further work must be performed to explore if: 1) the hypotensive effect of acute CBG is present in humans, 2) if the hypotensive effect is sustained after chronic use and 3) if the agonist effect is creating a sedative or anxiolytic effect.
Comment 2: I am not convinced why the authors increased the dose to 15 mg/kg i.p. in females. The reference 10 reports that CBG (10 and 20 mg/kg) significantly reduced mechanical hypersensitivity in neuropathic female mice with the 20 mg/kg dose of CBG showing the greatest significant difference compared to vehicle controls. Why the 20 mg/kg dose did not used in the paper?
Response to Comment 2: Thank you for bringing up this insightful question about dosing. While our previous work showed efficacy at 20 mg/kg for females in an acute scenario, this project aimed to identify if a lower dose, 15 mg/kg, may reduce pain in a long-lasting manner if administered daily. There is potential that a higher dose, such as 20 mg/kg, may induce a higher antinociceptive effect; however, higher doses also have the potential to induce additional side effects. We acknowledge that a dose-response curve for this chronic treatment was not performed and are planning future work to identify the scale of analgesia produced by different doses of daily CBG administrations. We have added this to last paragraph in Discussion, on limitations, as follows:
A major limitation of this work is the lack of a dose-response model for males and females receiving daily doses of CBG. Future work addressing this pharmacological question may reveal additional insight into minimum effective doses and sex differences in responses.
Minor reviews (comments 3-6)
Comment 3- Replace 10mg/kg and 15 mg/kg to 10 mg/kg and 15 mg/kg
Response to Comment 3: Thank you for this grammatical clarification, it has been addressed throughout the manuscript.
Comment 4- Line 184 are is duplicated
Response to Comment 4: Thank you for this identification of oversight, it has been fixed in line 184.
5- Figure 2 might be replaced by a table showinf initial weight (before treatment), 7 and 14 days
Response to Comment 5: Thank you for this suggestion, we have replaced Figure 2 with a table of initial/7 day/14 day time points, and added the graph to supplemental materials.
6 - In the Abstract, replace analgesic relief to antinociceptive effect
Response to Comment 6: Thank you for this edit, it has been addressed in the abstract.
Reviewer 2 Report
The manuscript describes the investigation of the ability of chronic CBG to alleviate mechanical hypersensitivity due to CIPN in mice by measuring responses to the analgesic effects of 7 and 14 days of daily CBG treatment.
The methods are appropriate with adequate reproducibility; results are informative; discussion is thorough.
Thus it can be accepted after minor revision.
The introduction is well laid-out. Recommend re-writing lines 55-57 which appears repetitive. Would also recommend adding works on CBG’s conformational shifts and drug-drug interactions to emphasize its pharmacodynamic profile: doi 10.1039/d3ob00383c rsc.li/obc
Do the authors have a working hypothesis as to why female mice seem less responsive to CBG given that there is no trend between administration and the estrous cycle? If so there this is not clear in the text.
Rewrite Line27-28.
The main results should be stated clearly in the Conclusion.
Fig. 2 must be replotted, with adjusted y-scale to reduce blank space.
minor editing
Author Response
Thank you very much for the kind words and important feedback to this manuscript. I have uploaded a track changes version of the manuscript to address these changes, and the steps we have taken are outlined in this response.
Comment 1: The introduction is well laid-out. Recommend re-writing lines 55-57 which appears repetitive. Would also recommend adding works on CBG’s conformational shifts and drug-drug interactions to emphasize its pharmacodynamic profile: doi 10.1039/d3ob00383c rsc.li/obc
Response to Comment 1: Thank you for this feedback. Lines 55-57 were repetitive with the preceding lines 50-55. Therefore, lines 50-57 were adjusted to reduce clutter and repetition. The new text is as follows:
There is growing patient and research interest in cannabinoid formulations with Δ9-tetrahydrocannabinol (THC) for pain syndromes, in part due to wide pharmacological activity with relatively limited side effects. While these pain therapeutics appear effective, recent clinical data suggest that patients may increase use over time to maintain analgesic efficacy, raising concerns for tolerance and abuse liability [5].
Novel non-euphoric cannabinoids, like cannabigerol (CBG), are concurrently gaining traction due to improved synthesis methods and changing public perception [6]. In fact, many recent studies of CBG in rodent models as well as human survey data suggest an increasing interest in using the compound to address human disease [7-12].
Additionally, thank you for the suggestion of the manuscript by Salha and colleagues, it has been added as citation 12.
Comment 2: Do the authors have a working hypothesis as to why female mice seem less responsive to CBG given that there is no trend between administration and the estrous cycle? If so there this is not clear in the text.
Response to Comment 2: Thank you for this insightful feedback. Lines 267-274 include potential explanations to differential responses to CBG analgesia. It reads:
While responses in female mice were not related to estrous cycle phase, the pharmacological sex differences in response are yet to be explained. Several mechanisms may contribute to differential response including sex differences in CBG metabolism, effects of sex hormones, and differential receptor signaling and density. Our future work includes pharmacokinetic analysis of metabolic rates and a more thorough understanding of sex hormone effect on analgesic efficacy by correlating serum sex hormone levels to analgesic effect.
Comment 3: Rewrite Line27-28.
Response to Comment 3: Thank you, the lines 27-28 now state:
These findings support the study of CBG as a long-lasting neuropathic pain therapy, which acts without tolerance in both males and females.
Comment 4: The main results should be stated clearly in the Conclusion.
Response to Comment 4: Thank you, there is now a Conclusion section (5.0), lines 436-446, which states the main results.
Comment 5: Fig. 2 must be replotted, with adjusted y-scale to reduce blank space.
Response to Comment 5: Thank you, Figure 2 has been replotted and re-uploaded to Supplemental Data. Figure 2 in the manuscript was replaced with a Table demonstrating mean and standard deviation weights of mice at start, after 7 days, and after 14 days of injections after a suggestion from another reviewer.
Reviewer 3 Report
In the present study, Rahul Nachnani et al investigated the chronic effects of cannabigerol in a mouse model of peripheral neuropathy induced by cisplatin. The study is well-designed and well-written. However, I have a few suggestions that could potentially enhance the current manuscript. These include improving clarity, highlighting novelty, and providing more detailed limitations. Below is the details
(1) The present study lacks novelty, as the efficacy of CBG has already been published in a prior study conducted by the same research group (https://doi.org/10.1002/ejp.2016). The main point of differentiation between the two studies lies in the duration of drug effects. However, I recommend that the authors highlight the unique aspects of the present study, which were not covered in the previous publication, such as the absence of analgesic tolerance and the consideration of factors like the estrous cycle. This distinction will assist readers in differentiating between the two studies.
(2) To enhance reader comprehension, it would be beneficial for the authors to include the number of mice used per experimental group in each figure legend.
(3) The authors should also provide a comprehensive discussion of the limitations of the current study. For example, the use of only one dose precludes the establishment of a dose-response relationship. Additionally, the study should address limitations related to behavioral pain testing and any other potential constraints that may affect the interpretation of the results.
Author Response
Thank you very much for the kind words and important feedback to this manuscript. I have uploaded a track changes version of the manuscript to address these changes, and the steps we have taken are outlined in this response.
Comment 1: The present study lacks novelty, as the efficacy of CBG has already been published in a prior study conducted by the same research group (https://doi.org/10.1002/ejp.2016). The main point of differentiation between the two studies lies in the duration of drug effects. However, I recommend that the authors highlight the unique aspects of the present study, which were not covered in the previous publication, such as the absence of analgesic tolerance and the consideration of factors like the estrous cycle. This distinction will assist readers in differentiating between the two studies.
Response to Comment 1: Thank you for this feedback. The first sentence of the discussion has been adjusted to highlight these differences, it reads as follows (lines 219-222):
We herein demonstrate that CBG reduces neuropathic pain in a mouse model of cisplatin-induced peripheral neuropathy (CIPN) in male and female mice, without development of tolerance or need for dosing more than once a day, regardless of estrous cycle phase.
Comment 2: To enhance reader comprehension, it would be beneficial for the authors to include the number of mice used per experimental group in each figure legend.
Response to Comment 2: Thank you for this suggestion, Figures 2-6 now have the mice used per experimental group in the figure legends and figure captions.
Comment 3: The authors should also provide a comprehensive discussion of the limitations of the current study. For example, the use of only one dose precludes the establishment of a dose-response relationship. Additionally, the study should address limitations related to behavioral pain testing and any other potential constraints that may affect the interpretation of the results.
Response to Comment 3: Thank you for this suggestion, we have added a dedicated limitations paragraph to the end of Discussion, printed here (lines 290-312).
A major limitation of this work is the lack of a dose-response model for males and females receiving daily doses of CBG. Future work addressing this pharmacological question may reveal additional insight into minimum effective doses and sex differences in responses. Then, the pharmacodynamic profile of CBG is not fully understood, such as its potential interactions with subtypes of the alpha-2 receptor and downstream behavioral effects of alpha-2 activation. Further work must be performed to explore if: 1) the hypotensive effect of acute CBG is present in humans, 2) if the hypotensive effect is sustained after chronic use and 3) if the agonist effect is creating a sedative or anxiolytic effect which confounds antinociceptive effects. The alpha-2 receptor driven hypotensive effect of CBG may be similar to that of the clinically used alpha-2 receptor agonist clonidine, which is effective at reducing blood pressure for emergent and urgent hypertensive crises, but has little-to-no efficacy at reducing blood pressure long term [22]. Additionally, this model of neuropathy only utilizes cisplatin, and other chemotherapy-induced neuropathy models may yield additional insights into the antinociceptive effects of CBG. Finally, the majority of non-significant results of our transcriptional inquiries were surprising but rational considering the timespan of mRNA and transcriptional changes. Our capturing of gene expression at this stage of the neuropathy and treatment modality (24 hours after the last drug injection, 3 weeks after the last cisplatin injection) may not be ideal for transient gene expression changes, despite the marked behavioral difference. Moreover, there is no compelling literature suggesting that pharmacological analgesia should produce changes in gene expression. Our future work will consider protein-level changes as well as circulating endocannabinoids as potential mechanistic contributions to the long-lasting analgesia induce by CBG.
Round 2
Reviewer 1 Report
The manuscript has improved with corrections.
Reviewer 3 Report
All the comments have been adressed by authors.